# Cost effective active search

**Shali Jiang**
CSE, WUSTL
St. Louis, MO 63130
jiang.s@wustl.edu

**Benjamin Moseley**
Tepper School of Business, CMU and
Relational AI
Pittsburgh, PA 15213
moseleyb@andrew.cmu.edu

**Roman Garnett**
CSE, WUSTL
St. Louis, MO 63130
garnett@wustl.edu

## Abstract

We study a paradigm of active learning we call *cost effective active search,* where the goal is to find a given number of positive points from a large unlabeled pool with minimum labeling cost. Most existing methods solve this problem heuristically, and few theoretical results have been established. Here we adopt a principled Bayesian approach for the first time. We first derive the Bayesian optimal policy and establish a strong hardness result: the optimal policy is hard to approximate, with the best-possible approximation ratio bounded below by $\Omega(n^{0.16})$. We then propose an *efficient and nonmyopic* policy, simulating future search progress using the negative Poisson binomial distribution. We propose simple and fast approximations for its expectation, which serves as an essential role in our proposed policy. We conduct comprehensive experiments on drug and materials discovery datasets and demonstrate that our proposed method is superior to a popular (greedy) baseline.

## 1 Introduction

Active search is a particular realization of active learning where the goal is to identify positive points from a large pool of unlabeled candidates. To make this problem copelling, we typically assume that the positive points are extremely rare (e.g., $<1\%$) and that labeling process costly, making the search challenging. A prototypical application is drug discovery, where we seek to identify compounds exhibiting binding activity with a certain biological target from among millions of candidates [Warmuth et al., 2001, Garnett et al., 2015]. Other applications include materials discovery [Jiang et al., 2018b], product recommendation [Vanchinathan et al., 2015], etc.

There have been major theoretical and algorithmic advances on active search in recent years [Garnett et al., 2012, 2015, Jiang et al., 2017, 2018a]. This work has primarily focussed on the budgeted setting, where the goal is to find as many positive points as possible in a given budget $B$, the total number of points one can query, known *a priori*. In this paper, we consider the "dual" problem: *how to find a given number of positives with minimum cost*. Formally, given a large pool of $n$ points $\mathcal{X} = \{x_i\}_{i=1}^n$ with corresponding unknown labels $y_i \in \{0, 1\}$ indicating whether $x_i$ is positive or not, we want to sequentially choose a set $\mathcal{D} \subset \mathcal{X}$ of points to evaluate to identify a given target number $T$ of positives in as few iterations as possible. We call this problem *cost effective active search* (CEAS). To contrast with the budgeted setting, we present both formulations:

Budgeted: $\arg\max_{\mathcal{D} \subset \mathcal{X}} \sum_{x_i \in \mathcal{D}} y_i$, s.t. $|\mathcal{D}| = B$;   CEAS: $\arg\min_{\mathcal{D} \subseteq \mathcal{X}} |\mathcal{D}|$, s.t. $\sum_{x_i \in \mathcal{D}} y_i \geq T$.

One might wonder if we could reduce CEAS to the budgeted case, for which many effective policies are readily available [Jiang et al., 2017]. For example, given a CEAS problem with target $T$, we could instead solve a budgeted problem with an estimated budget $B$. However, this "dual" transformation is not one-to-one: given a budget $B$, the expected utility is no longer $T$. In fact, both $\Pr(\text{utility} \mid B)$ and $\Pr(\text{cost} \mid T)$ are highly complicated distributions and estimating their expectations is extremely intractable, as we will show later. Therefore it is necessary to develop specific policies for CEAS.

CEAS is a direct model of practical cases where we must achieve a certain target with minimum cost; for example, during initial screening for drug discovery, we may seek a certain number of active compounds to serve as lead compounds to be refined later in the discovery process. There have been several previous investigations into the the CEAS setting (e.g., Warmuth et al. [2001, 2003]), but the proposed policies are often based on heuristics, and to the best of our knowledge, few theoretical results have been established regarding the hardness of this problem.

In this paper, we study the CEAS problem under the Bayesian decision-theoretic framework. We first derive the Bayesian optimal policy and establish a strong hardness result showing that the optimal policy is extremely inapproximable. In particular, we show any efficient algorithm is at least $\Omega(n^{0.16})$ times worse than the optimal policy in terms of the expected cost needed.

We then propose nonmyopic approximations to the Bayesian optimal policy and develop efficient implementation and pruning techniques that make nonmyopic search in large spaces possible. To that end, we discuss an understudied distribution called the "negative Poisson binomial" and propose a simple and fast approximation to its expectation, an essential component of our efficient nonmyopic policy. We conduct comprehensive experiments on benchmark datasets in various domains such as drug and materials discovery and demonstrate that our policy is superior to widely used baselines.

## 2 Related work

Active search in the budgeted setting has been well studied under the Bayesian decision-theoretic framework in recent years. Garnett et al. [2012] derived the Bayesian optimal policy and proved that an $\ell$-step lookahead policy can be arbitrarily better than an $m$-step one for any $\ell > m$ and proposed the use of myopic approximations in practice, e.g., one- or two-step lookahead, for computational tractability. Garnett et al. [2015] comprehensively compared one- and two-step lookahead policies on 360 drug discovery datasets and showed that the two-step policy performs significantly better than greedy search most of the time.

Jiang et al. [2017] proposed an efficient nonmyopic approximation to the Bayesian optimal policy for the budgeted setting by assuming conditional independence and generalized this approach to batch policies [Jiang et al., 2018a]. They also proved a hardness result showing that the approximation ratio for the budgeted case is bounded above by $O(1/\sqrt{\log n})$ (note it is a maximization setting). Here we apply a similar proof technique to the CEAS setting but prove a much stronger bound of order $\Omega(n^{0.16})$.

Active search is a form of active learning with a special utility (or cost) function. Chen and Krause [2013] studied minimum-cost active learning under the version space reduction utility and proved that a greedy algorithm is near-optimal due to adaptive submodularity. Warmuth et al. [2001] studied "active learning in the drug discovery process." They used perceptron and SVM as predictive models and tried several variants of uncertainty sampling (sampling points closest to the separating hyperplane) and greedy sampling (choosing points furthest from the hyperplane). The authors found that greedy sampling performed better, but offered no theoretical justification, in contrast to our decision theoretic approach.

CEAS is a special case of the *adaptive stochastic minimum cost cover* problem [Golovin and Krause, 2011], for which an inapproximability bound was proved using a spiritually similar instance construction to our own, including the idea of "treasure hunting" and using XOR for encoding. However, their worst case construction is much more complicated and does not directly correspond to active search. Furthermore, their bound only holds conditionally (PH $\neq \Sigma_2^P$). Our proof is much simpler and does not rely on any complexity theoretic hypothesis.

Active search is also closely related to the *total recall* problem in information retrieval [Grossman et al., 2016, Yu and Menzies, 2018, Renders, 2018] or the so-called technology-assisted review progress for evidence discovery in legal settings [Grossman and Cormack, 2010], where the goal is to retrieve (nearly) *all* relevant documents at a reasonable labeling cost. Despite the similarity to our setting, retrieving *all* positives could be drastically different (and arguably much harder). One immediate difference is that we often do not know the total number of positives in a dataset *a priori,* thus a considerable amount of research is devoted to deciding when to stop in the total recall procedure [Grossman et al., 2016]. Our definition of CEAS avoids this complication.

# 3 Bayesian optimal policy

We motivate our approach using Bayesian decision theory. We assume there is a model that provides the posterior probability of a point $x$ being positive, conditioned on previously observed data $\mathcal{D}$, i.e., $\Pr(y = 1 \mid x, \mathcal{D})$. The Bayesian optimal policy then chooses the point that minimizes the expected number of further iterations required. Formally, in iteration $(i + 1)$, where we have observed $\mathcal{D}_i$, then

$$x^* = \arg\min_x \mathbb{E}[c^r \mid x, \mathcal{D}_i], \tag{1}$$

where $c^r$ denotes the further cost incurred (i.e., the size of the chosen set) when $r$ more positives are identified after $\mathcal{D}_i$, and $r = T - \sum_{(x,y) \in \mathcal{D}_i} y$ is the remaining target yet to be achieved.

The expected cost can be written as a Bellman equation:

$$\mathbb{E}[c^r \mid x, \mathcal{D}_i] = 1 + \Pr(y = 1 \mid x, \mathcal{D}_i) \cdot \min_{x'} \mathbb{E}[c^{r-1} \mid x', x, y = 1, \mathcal{D}_i] + \\ \Pr(y = 0 \mid x, \mathcal{D}_i) \cdot \min_{x'} \mathbb{E}[c^r \mid x', x, y = 0, \mathcal{D}_i]. \tag{2}$$

However, this recursion is not mathematically well defined since there is no exit. To see this, consider the base case where $r = 1$, i.e., only one more positive left to find. With probability $\Pr(y = 1 \mid x, \mathcal{D}_i)$, $x$ is positive, and we finish with cost 1; with probability $1 - \Pr(y = 1 \mid x, \mathcal{D}_i)$, $x$ is negative, and then we need to update the probability conditioned on $y = 0$ and repeat this process. That is,

$$\mathbb{E}[c^1 \mid x, \mathcal{D}_i] = \Pr(y = 1 \mid x, \mathcal{D}_i) \cdot 1 + (1 - \Pr(y = 1 \mid x, \mathcal{D}_i)) \cdot \min \mathbb{E}[c^1 \mid x, \mathcal{D}_i]. \tag{3}$$

So even for $r = 1$, the recursion never exits. Note this is very different from the budgeted setting [Garnett et al., 2012], where the base case with budget 1 is well defined and trivial to compute.

In practice, the recursion must stop after exhausting the whole pool. We assume the search stops after at most $t$ steps; we stress that this is only for derivation purposes, and should not be confused with the budgeted setting with budget $t$. If we set $t = |\mathcal{X}|$, then that means we keep searching until the target is achieved or there are no more points left to evaluate. Let $c_t^r$ be the cost incurred after $r$ positives are found or $t$ evaluations are completed. Now we can derive the base case of the recursion. To simplify notations, we will omit the conditioning on $\mathcal{D}_i$ in the following and assume all probabilities are implicitly conditioned on current observations. Let $p(x) = \Pr(y = 1 \mid x, \mathcal{D}_i)$. We start with the simplest case $r = 1$:

When $t = 1$, the expected cost would be 1 no matter what.

When $t = 2$, $\mathbb{E}[c_2^1 \mid x] = p(x) \cdot 1 + (1 - p(x)) \cdot 2 = 2 - p(x)$. In this case, the greedy policy choosing the point with highest probability is optimal.

When $t = 3$,

$$\mathbb{E}[c_3^1 \mid x] = p(x) \cdot 1 + (1 - p(x)) \cdot \min_{x'} \mathbb{E}[c_2^1 \mid x', x, y = 0] \\ = 2 - p(x) - (1 - p(x))\max_{x'} p'(x'). \tag{4}$$

where $p'(x') = \Pr(y' = 1 \mid x', x, y = 0)$. We can see

$$\arg\min_x \mathbb{E}[c_3^1 \mid x] \Leftrightarrow \arg\max_x p(x) + (1 - p(x))\max_{x'} p'(x'). \tag{5}$$

Note the form on the right hand side of (5) has a clear exploration vs. exploitation explanation. It balances between choosing a point with high probability as governed by $p(x)$ (exploit) and a point that has low probability but would lead to a high probability point if it turns out to be negative as governed by $(1 - p(x)) \max_{x'} p'(x')$ (explore). This is counter-intuitive since one would imagine with only one positive left to find, the best thing to do should be greedy, but we have shown if we are granted three or more iterations, greedy might not be optimal in terms of minimizing expected cost. Therefore, we argue that nonmyopic planing is crucial for CEAS.

We draw a connection between (5) and the two-step score [Garnett et al., 2012] in budgeted setting:

$$\arg\max_x p(x) + p(x)\max_{x'} p(y' = 1 \mid x', y = 1) + (1 - p(x))\max_{x'} p(y' = 1 \mid x', y = 0); \tag{6}$$

we can see in (6), the future utility is averaged over the positive and negative case, whereas in (5) only the negative case is considered.

The recursion is now well-defined for $r = 1$:

$$\mathbb{E}[c_t^1 \mid x] = p(x) \cdot 1 + (1 - p(x)) \cdot \min_{x'} \mathbb{E}[c_{t-1}^1 \mid x', x, y = 0]. \tag{7}$$

Note it takes $\mathcal{O}(n^t)$ time to compute even for the simplest case $r = 1$. In general $(r \geq 1)$, we have

$$\mathbb{E}[c_t^r \mid x] = 1 + p(x) \cdot \min_{x'} \mathbb{E}[c_{t-1}^{r-1} \mid x', x, y = 1] +$$
$$(1 - p(x)) \cdot \min_{x'} \mathbb{E}[c_{t-1}^r \mid x', x, y = 0]. \quad (8)$$

### 3.1 Hardness of cost effective active search

The above derivation indicates that we can not efficiently compute the expected cost exactly. In fact, the optimal policy is not only hard to compute, it is even hard to approximate. As we show in the following theorem, any efficient algorithm is at least $\Omega(n^{0.16})$ times worse than the optimal policy in terms of average cost:

**Theorem 1.** *Any algorithm $\mathcal{A}$ with computational complexity $o\left(n^{n^\epsilon}\right)$ has an approximation ratio $\Omega(n^\epsilon)$, for $\epsilon = 0.16$; that is,*

$$\frac{\mathbb{E}[\text{cost}_{\mathcal{A}}]}{\text{OPT}} = \Omega\left(n^\epsilon\right), \quad (9)$$

*where $\mathbb{E}[\text{cost}_{\mathcal{A}}]$ is the expected cost of $\mathcal{A}$, and* OPT *is that of the optimal policy.*

Note that this lower bound is very tight since $\mathcal{O}(n)$ is a trivial upper bound. We prove the theorem by constructing a class of active search instances similar to that of the hardness proof for the budgeted setting [Jiang et al., 2017]. Specifically, there is a small secret set of points, the labels of which encode the location of a clump of positive points. The optimal policy (with unlimited computational power) could easily identify this secret set by enumerating all possible subsets of the same size and feeding them into the inference model, thereby revealing the positive clump and completing the search quickly. However, for an algorithm with limited computational power, the probability of revealing this secret set is extremely low, and hence it can not do any better than randomly guessing, which results in a much higher expected cost.

Our proof has two key differences compared to previous work. First, it results in an *exponentially* stronger bound ($n^{0.16}$ vs $\sqrt{\log n}$). Second, the improved bound requires a different methodology on how to hide the set of profitable points from the algorithm. In particular, a new counting technique is used for bounding the probability of identifying the "secret set". In the budgeted setting, one key argument considered "If an efficient policy selects a subset of $B$ points, how likely is it to identify the secret set?", which was straightforward to compute since the chosen budget $B$ defined the cardinality of the selected subset. Such reasoning does not apply to CEAS setting. In fact, we leverage the underlying algorithmic challenge of optimizing the newly considered objective where the algorithm has to continue searching until it reaches the target. A formal proof is given in the appendix.

## 4 Approximation of the Bayesian optimal policy

The main cause of the high complexity for computing the expected cost is that we need to recursively update the probabilities of the remaining points conditioned on possible outcomes of the previous points. So a natural way to relieve the burden is to assume *conditional independence* (CI) after several steps; e.g., after observing the point we are considering. This idea has been adopted to approximate the expected utility in the budgeted setting, and was demonstrated to perform very well [Jiang et al., 2017, 2018a] in practice. We propose to adapt this idea to CEAS. While in the budgeted setting, the expected future utility under the CI assumption is simply the sum of the top probabilities up to the remaining budget, it is much less straightforward to compute the expected cost in CEAS setting.

We model the remaining cost as a *negative Poisson binomial* (NPB) distribution, i.e., the number of coins (with nonuniform biases) that need to be flipped (independently) to get a given number of HEADs. This is in contrast to the utility being modeled as a *Poisson binomial* (PB) distribution in the budgeted setting. The NPB distribution is a natural generalization of the *negative binomial* (NB) distribution where all coins have the same biases. While both NB and PB distributions are well studied, few references can be found for NPB. Some informal discussions about NPB can be found in the Physics Forum[1]. [Charalambous, 2014] defines a distribution they also call NPB, but it is actually a sum of geometric variables. [Liebscher and Kirschstein, 2017] defines NPB as the number of "failures given successes" for predicting the outcome of darts tournaments. In the following, we formally

define the NPB distribution and propose novel and fast approximations to its expectation, based on which we derive our efficient nonmyopic policy.

### 4.1 Negative Poisson binomial distribution

We define the NPB distribution in an intuitive manner as follows:

**Definition 1** (Negative Poisson binomial distribution). *Let there be an infinite number of ordered coins with HEADs probabilities $p_1, p_2, p_3, \cdots$; given the number $r$ of HEADs required, we toss the coins one by one in this order until $r$ HEADs occur. We say the number of coins tossed $m$ follows a negative Poisson binomial distribution, or $m \backsim \text{NPB}(r, [p_1, p_2, p_3, \cdots])$.*

Note that this distribution is supported for any integer $m \geq r$. We adopt a truncated version where we have a finite number of coins, $n$, and we assume $n$ is large enough so that $\Pr(m \geq n)$ is negligible, which is typically the case in active search since $r \ll n$, where $n$ the size of unlabeled pool and $r$ is the target to achieve.

Its PMF can be derived using the PMF of a Poisson binomial (PB) distribution $r \backsim \text{PB}([p_1, p_2, \cdots, p_n])$, which is the number of HEADs if we independently toss $n$ coins with HEADs probabilities $p_1, p_2, \ldots, p_n$. Denote $\Pr_{\text{PB}}(i, j)$ as the probability of $j$ HEADs when there are $i$ coins with probabilities $p_1, p_2, \ldots, p_i$, then $\Pr_{\text{PB}}(n, r)$ can be computed via dynamic programming (DP):

$$\Pr_{\text{PB}}(n, r) = \begin{cases} \prod_{i=1}^{n}(1 - p_i), & \text{if } r = 0; \\ p_n \Pr_{\text{PB}}(n-1, r-1) + (1 - p_n)\Pr_{\text{PB}}(n-1, r), & \text{if } 0 < r <= n; \end{cases} \tag{10}$$

The DP table of $\Pr_{\text{PB}}(n, r)$ is of size $\mathcal{O}(nr)$. Chen and Liu [1997] also derived other formulas for computing the PMF of the PB distribution.

Given the PMF of the PB distribution, the PMF of $m \backsim \text{NPB}(r, [p_1, \cdots, p_n])$ is: $\forall\, m \geq r$

$$\Pr_{\text{NPB}}(m, r) = p_m \Pr_{\text{PB}}(m-1, r-1), \tag{11}$$

and the expectation is $\mathbb{E}[m] = \sum_{i=r}^{n} \Pr_{\text{NPB}}(i, r) \cdot i$. Note in computing $\Pr_{\text{NPB}}(n, r)$, all $\Pr_{\text{NPB}}(m, r)$ for $m < n$ are also computed. So the complexity for computing the expectation is also $\mathcal{O}(nr)$.

### 4.2 Approximation of the NPB expectation

The complexity $\mathcal{O}(nr)$ for computing the expectation is prohibitively high. To reduce its complexity, we observe in practice $\Pr_{\text{NPB}}(m, r)$ will be very close to zero for $m \gg r$. So we can stop at $\bar{m}$ when $\sum_{i=r}^{\bar{m}} \Pr_{\text{NPB}}(i, r) \geq 1 - \epsilon$ for, e.g., $\epsilon = 10^{-6}$. Hence, an almost exact solution can be computed in $\mathcal{O}(\bar{m}r)$. We call this $\epsilon$-DP.

The complexity $\mathcal{O}(\bar{m}r)$ for computing the expectation in (4.1) may still be high since we might need to compute this expectation for every candidate point in each iteration. We propose another cheap but accurate approximate method with only $\mathcal{O}(\mathbb{E}[m])$ complexity. The idea is simple: coin toss $i$ contributes $p_i$ HEADs in expectation, so we accumulate this until $r$ HEADs occur. That is,

$$\mathbb{E}[m] \approx \arg\min_k \sum_{i=1}^{k} p_i \geq r. \tag{12}$$

We call this approximation ACCU (short for "accumulate"). Note that ACCU always returns an integer, while the true expectation might not be integral. To fix this, we subtract a correction term. Let $\hat{m} = \arg\min_k \sum_{i=1}^{k} p_i \geq r$. We check what portion of $p_{\hat{m}}$ was needed for the sum to be exactly $r$, and remove the extra portion. That is,

$$\mathbb{E}[m] \approx \hat{m} - \frac{\left(\sum_{i=1}^{\hat{m}} p_i\right) - r}{p_{\hat{m}}}. \tag{13}$$

We call this approximation ACCU'. In the special case of the negative binomial distribution (i.e. $p = p_1 = p_2 = \cdots = p_n$), ACCU' recovers the true expectation $r/p$.

One might be tempted to approximate the expectation by a natural generalization of the expectation of a negative binomial distribution $r/p = 1/p + 1/p + \cdots + 1/p$; that is, $\mathbb{E}[m] \approx 1/p_1 + 1/p_2 + \cdots + 1/p_r$. We call this RECIP. Note that this approximation is also exact when $p_1 = p_2 = \cdots = p_n = p$. We will see that this approximation can be very poor.

Table 1: Time cost and quality of various approximations of the expectation of NPB distribution.

|  | EXACT | $\epsilon$-DP | ACCU' | ACCU | RECIP |
|---|---|---|---|---|---|
| RMSE | - | 0.0004 | 0.0438 | 0.5723 | 7.8276 |
| time(s) | 97.4841 | 0.1851 | 0.0029 | 0.0029 | 0.0002 |

We run simulations to demonstrate how close these approximations are. We parameterize the NPB distribution using the posterior marginal probabilities $[p_1, \ldots, p_n]$ computed in a typical active search iteration using a $k$-nn model (see Figure 2 in the appendix). We plot the approximation errors against the EXACT expectation computed via (4.1) for $r = 1, \ldots, 500$, shown in Figure 1a. We can see $\epsilon$-DP has basically zero error everywhere. ACCU' also has zero error almost everywhere except two locations. ACCU constantly overestimates the exact value by a small fraction, due to being an integer, whereas RECIP considerably underestimates the true value, especially when $r$ is large. The root mean square error (RMSE) and total time cost for computing the 500 expectations are shown in Table 1.

After a closer look at the locations where the errors are higher (e.g. around $r = 220$, for which $\mathbb{E}[m] \approx 300$), we find that such locations exactly correspond to $r$ values such that the probability around $\mathbb{E}[m]$ drops abruptly (refer to Figure 2 in the appendix). This makes perfect sense since ACCU' and RECIP are not aware of such changes after $\mathbb{E}[m]$. RECIP suffers from this problem more severely since it only looks at $r$ probability values but ACCU' looks at $\mathbb{E}[m]$ values. Overall, we can see ACCU' serves as an appropriate time-quality tradeoff, and we will use this approximation for our policy. It is an interesting question whether we can derive error bounds for ACCU'.

## 4.3 Efficient nonmyopic approximations

We can approximate the expected cost (8) as follows:

$$\mathbb{E}[c^r \mid x, \mathcal{D}_i] \approx 1 + \Pr(y = 1 \mid x, \mathcal{D}_i)\mathbb{E}[m^+] + \Pr(y = 0 \mid x, \mathcal{D}_i)\mathbb{E}[m^-] \equiv f(x), \qquad (14)$$

where $\mathbb{E}[m^+] \approx \mathbb{E}[c^{r-1} \mid x, y = 1, \mathcal{D}_i]$, $\mathbb{E}[m^-] \approx \mathbb{E}[c^r \mid x, y = 0, \mathcal{D}_i]$, and $\Pr(c^{r-1} \mid x, y = 1, \mathcal{D}_i)$ and $\Pr(c^r \mid x, y = 1, \mathcal{D}_i)$ are assumed to be NPB distributions defined by posterior marginal probabilities (in descending order) after the respective conditioning . We use ACCU' to compute the approximate expectations. In every iteration, we choose $x$ minimizing $f(x)$. We will call this policy efficient nonmyopic cost effective search, or ENCES.

We further propose to adapt our policy by treating the remaining target $r$ as a tuning parameter. In particular, we argue that it is better to set this parameter smaller than the actual remaining target. The rationale is three-fold: (1) the approximation based on the NPB distribution typically over estimates the actual expected cost, since we are pretending the probabilities will not change after the current iteration, whereas actually the top probabilities defining the NPB distribution will increase considerably as we discover more positives. So the actual expected cost should be much smaller. (2) even if the CI assumption is correct, planning too far ahead might hurt if the model is not accurate enough; after all, "all models are wrong." (3) setting $r$ smaller makes the bounds on the expectation much tighter, hence the pruning is much more effective (details on pruning included in the appendix).

We consider two schemes: setting $r$ to a constant or proportional to the remaining target. For example, ENCES-10 means we always set $r = 10$ if the remaining target is greater than 10, otherwise we use the actual remaining target; and ENCES-0.2 means we set $r$ to be 20% of the actual remaining target.

## 5 Experiments

We conduct experiments to compare our proposed policy against three baselines[2]:

GREEDY: this is the most widely used policy, which always chooses a point with the highest probability. It is an equivalent of choosing the point furthest from the hyperplane in the case of an SVM model [Warmuth et al., 2001]. It has been shown that this baseline is hard to beat in the total recall setting [Grossman et al., 2016].

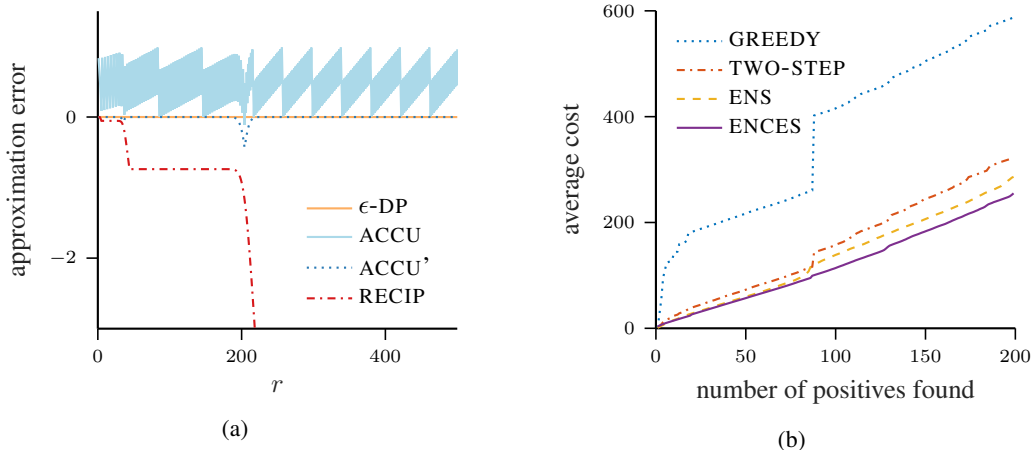

Figure 1: (a) Approximation errors of various approximation methods for computing the expectation of NPB distribution. $y$-axis is approximate $\mathbb{E}[m]$ minus exact $\mathbb{E}[m]$. The error for RECIP drops below -10 after about $r = 250$, hence not shown to zoom in the interesting part of the figure. (b) Average cost versus the number of positives found, averaged over 9 drug discovery datasets.

Table 2: Results on materials discovery data. Averaged over 30 experiments.

| | 50 | 100 | 200 | 300 | 400 | 500 | 1000 | 1500 | average |
|---|---|---|---|---|---|---|---|---|---|
| GREEDY | 84.5 | 175.0 | 347.7 | 522.5 | 721.8 | 924.1 | 2025.9 | 2981.7 | 972.9 |
| TWO-STEP | 86.0 | 179.1 | 349.0 | 533.2 | 735.0 | 938.1 | 1973.1 | 3019.4 | 976.6 |
| ENS-70 | *81.5* | *165.1* | *337.7* | 524.6 | 720.0 | *910.0* | 1790.6 | *2757.0* | 910.8 |
| ENS-0.7 | **80.2** | 167.8 | **328.4** | **509.7** | 708.6 | *887.4* | 1798.5 | *2773.6* | 906.8 |
| ENCES-50 | 85.2 | **156.4** | *330.8* | *518.4* | *701.4* | **885.7** | **1745.4** | *2753.2* | *897.1* |
| ENCES-0.5 | 87.1 | 164.3 | *336.9* | *513.7* | **683.9** | *891.9* | *1774.1* | **2724.0** | **897.0** |

TWO-STEP: the budgeted two step lookahead policy as defined in (6). Note we use (6) instead of (5) since (5) looks ahead by considering the maximum probability a point would lead to if it is negative, so it does not make much sense to use this policy with $k$-nn, which only models positive correlation.

ENS: the efficient nonmyopic search policy recently developed by Jiang et al. [2017] for the budgeted setting, informally defined as follows: $\arg\max_x p(x) + \mathbb{E}_y[\text{sum of top } b \text{ prob.} \mid x, y]$, where $b$ is the remaining budget after choosing $x$. This policy was shown to perform remarkably well in various domains due to its adaptation to the remaining budget. However, to apply it in the CEAS setting, we have to make a simple modification since these is no concept of budget. We also experiment with two schemes of setting $b$: constant or proportional to the remaining target. We test ENS-10,30,50,70 and ENS-0.1, 0.3, 0.5, 0.7, and compare our method against the best one. Such $b$'s underestimate the true budget, but the same rationale given in Sec. 4.3 applies.

We run our policies with $r = 10, 20, 30, 40, 50$ and $0.1, 0.2, 0.3, 0.4, 0.5$, and also report the results of the best one (on average) in each scheme. Full results are in the appendix. We report results on drug and materials discovery data [Jiang et al., 2017]. In all our experiments, we start the search with a single randomly selected positive point, and repeat the experiments 30 times.

**Materials discovery**. We apply active search to discover novel alloys forming bulk metallic glasses (BMGs). The database [Jiang et al., 2017] is composed of 118 678 known alloys with 4 746 (about 4%) having the desired property. We test $T = 50, 100, 200, 300, 400, 500, 1000, 1500$. Table 2 shows the average cost. Following Jiang et al. [2018a], we highlight the entries with lowest cost in boldface, and those not significantly worse than the best in blue *italic*, under one sided paired $t$-tests with $\alpha = 0.05$.

We summarize the results as follows: (1) all of the nonmyopic policies outperform GREEDY, and TWO-STEP performs on par with GREEDY. (2) ENCES variants are mostly the best or not significantly worse than the best. (3) ENS is a strong baseline, being the best or not significantly worse than ENCES

Table 3: Averaged results on nine drug discovery datasets, 30 experiments for each.

|  | 50 | 100 | 150 | 200 | average |
|---|---|---|---|---|---|
| GREEDY | 215.7 | 414.4 | 503.2 | 587.4 | 430.2 |
| TWO-STEP | 71.7 | 156.0 | 243.2 | 322.4 | 198.4 |
| ENS-30 | *58.8* | 134.9 | 208.3 | 283.3 | 171.3 |
| ENS-0.7 | *59.1* | 132.8 | 212.0 | 284.2 | 172.0 |
| ENCES-20 | **56.3** | **112.7** | **184.5** | **255.1** | **152.2** |
| ENCES-0.2 | *72.9* | 116.0 | 194.8 | 298.9 | 170.7 |

for several cases. (4) ENCES with $r$ being half of the remaining target performs the best on average, but $r = 50$ is not significantly worse.

**Drug discovery**. Now we consider the main application of active search: drug discovery. We simulate virtual drug screening to find chemical compounds exhibiting binding activities with a certain biological target. We use the first nine drug discovery datasets as described in Sec. 5.1 of Jiang et al. [2018a]. Each dataset corresponds to a different biological target. The number of positives in the nine datasets are 553, 378, 506, 1023, 218, 916, 1024, 431, 255, with a shared pool of $100\,000$ negatives. We set $T = 50, 100, 150, 200$. The average costs are shown in Table 3. Each entry in this table is averaged over the nine datasets and 30 experiments each, so in total 270 experiments for each policy and target $T$.

We see a consistent winner: ENCES-20. On average it outperforms all baselines by a large margin. In particular, it improves over GREEDY by a 56-73% reduction in average cost. ENCES-0.2 also performs very well, not significantly worse than ENCES-20 when $T = 50$. Though we only presented results of our method with the best parameters, we point out that it always outperforms GREEDY by a large margin for all other parameters. Full results are in the appendix. Also note that the tested ENS variants, which solve the CEAS problem by reducing it to the budgeted setting, are much better than the myopic methods, but still significantly worse than our proposed method, which suggests it is beneficial to design specific policies for cost effective setting.

To see how these policies behave during the course of search, we plot in Figure 1b the average cost for each policy along the way until $T = 200$ positives are found. The individual curves for each of the nine datasets are shown in the appendix. We only show ENCES-20 and ENS-30 in this plot; the curves of ENCES-0.2 and ENS-0.7 are very similar. We see the average cost of GREEDY exhibits a "piecewise linear" shape. This is likely due to its greedy behavior [Jiang et al., 2017]: it keeps exploiting the high probability points around a discovered positive neighborhood until they are exhausted, then it has to spend a long time to find another neighborhood in a blind-minded way since it did not learn much about the space, resulting in the discontinuity of the cost curve after a certain number of positives are collected. In contrast, TWO-STEP and ENS have much smoother behavior with minimal discontinuity, and ENCES has almost perfectly *linear cost* w.r.t. $T$, with the slope only slightly greater than one. This is a very desirable property for cost effective active search.

## 6 Conclusion

In this paper, we introduced and studied cost effective active search under the Bayesian decision-theoretic framework. This is the first principled study of the problem in the literature. We derived the Bayesian optimal policy and proved a novel hardness result: the optimal policy is extremely inapproximable, with approximation ratio bounded below by $\Omega(n^{0.16})$. We then proposed an efficient strategy to approximate the optimal policy using the negative Poisson binomial distribution and proposed efficient approximations for its expectation. We demonstrated the superior performance of our proposed policy to several baselines, including the widely used greedy policy and the state-of-the-art nonmyopic policy adapted from budgeted active search. The performance on drug discovery datasets was especially encouraging, with a 56–73% cost reduction on average compared to greedy sampling.

Regarding the tuning parameter in our policy, one rule-of-thumb is to set it to be relatively small (e.g., $\leq 50$ or $\leq 50\%$ of the remaining target). How to adapt it in a more principled way is an interesting

future direction. Another future direction is to extend the proposed method to batch setting, where multiple points are evaluated simultaneously.

## Acknowledgments

We would like to thank Mark Bober for providing support regarding computational services. SJ and RG were supported by the National Science Foundation (NSF) under award numbers IIA–1355406, IIS–1845434, and OAC–1940224. BM was supported by a Google Research Award and by the NSF under awards CCF–1830711, CCF–1824303, and CCF–1733873.

## Footnotes

[1]https://www.physicsforums.com/threads/negative-poisson-binomial-distribution.759630/

[2]Matlab implementations of our method and the baselines are available here: `https://github.com/shalijiang/efficient_nonmyopic_active_search.git`

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
