[Supplementary Material]

# Cost effective active search: supplementary materials

**Shali Jiang**
CSE, WUSTL
St. Louis, MO 63130
jiang.s@wustl.edu

**Benjamin Moseley**
Tepper School of Business, CMU and
Relational AI
Pittsburgh, PA 15213
moseleyb@andrew.cmu.edu

**Roman Garnett**
CSE, WUSTL
St. Louis, MO 63130
garnett@wustl.edu

## 1  Hardness of cost effective active search

In this section, we present the proof of Theorem 1 in the main text, restated below:

**Theorem 1.** *Any algorithm $\mathcal{A}$ with computational complexity $o\left(n^{n^\epsilon}\right)$ has an approximation ratio $\Omega(n^\epsilon)$, for $\epsilon = 0.16$; that is,*

$$\frac{\mathbb{E}[\text{cost}_\mathcal{A}]}{\text{OPT}} = \Omega\left(n^\epsilon\right), \tag{1}$$

*where $\mathbb{E}[\text{cost}_\mathcal{A}]$ is the average cost of $\mathcal{A}$, and* OPT *is that of the optimal policy.*

*Proof.* We begin our proof by constructing a very similar class of instances $\mathcal{H}$ as in Jiang et al. [2017], with different parameter settings. We reproduce the instance illustration in Jiang et al. [2017] here in Figure 1, and briefly summarize the essences; details can be found in the supplementary materials of Jiang et al. [2017]. Each instance in the class has $n$ points with two types: "clumps" and "isolated points".

**"Clumps."** As shown in Figure 1b, there are $n^{4\epsilon}$ clumps, and each one has $T = n^{2\epsilon}$ points with the same labels, where $\epsilon$ is a small constant such as $0.1$. There is exactly one positive clump. So *a priori* the marginal probability of each clump point being positive is $p_c = 1/n^{4\epsilon}$.

**"Isolated points."** The remaining $n - n^{6\epsilon} = \Theta(n)$ points are isolated points, they are all independent to each other; these points are further categorized into two classes: a "secret set" and totally independent points.

- The secret set, denoted by $\mathcal{S}$ (Figure 1a), encodes the location of the positive clump in the following way: $\mathcal{S}$ contains $m = \log_2(n^{4\epsilon}) = 4\epsilon \log n$ groups $\mathcal{S}_1, \mathcal{S}_2, \ldots, \mathcal{S}_m$, each of size $n^{2\epsilon}$; each $\mathcal{S}_i$ are further partitioned into $d = n^\epsilon/(1 - 5\epsilon)$ groups [1], with each group having $c = n^\epsilon$ points. Each of the $j$th ($j = 1, \ldots, d$ group in $\mathcal{S}_i, i = 1, \ldots, m$ encodes one virtual bit $b_{ij}$ by taking the OR operation on the $c$ labels (i.e. $b_{ij} = 1$ iff at least one of the $c$ points are positive); then the $d$ bits $b_{i1}, \ldots, b_{id}$ encode a virtual bit $b_i$ using XOR, i.e., $b_i = b_{i1} \oplus \cdots \oplus b_{id}$. We set probability of each point in $\mathcal{S}$ as $p_s = 1 - \frac{1}{2^{1/c}}$ so that $\Pr(b_{ij} = 1) = (1 - p_b)^c = \frac{1}{2}$, which also leads to $\Pr(b_i = 1) = \frac{1}{2}$. Finally the binary string $b_1 b_2 \cdots b_m$ determines the index of the positive clump.

- The remaining $n - T2^m - mdc$ points, denoted as set $\mathcal{R}$, are totally independent to each other and any other points. The probability of any point in $\mathcal{R}$ is also $p_s$.

The goal is to find $T$ points with minimum labeling cost.

The two observations given in the proof of Jiang et al. [2017] still hold, as restated as following.

(a) Secret set $\mathcal{S}$ of $mdc$ isolated points.    (b) $2^m T$ points from clumps.    (c) Isolated and independent points $\mathcal{R}$.

Figure 1: An instance of active search where any efficient algorithm can be arbitrarily worse than an optimal policy.

**Observation 1.** *At least $d$ points from $\mathcal{S}_i$ (for any $i$) need to be observed in order to infer one bit $b_i$ of information about the positive clump.*

**Observation 2.** *Observing any number of clump points does not change the marginal probability of any point in the secret set $\mathcal{S}$.*

Consider a random instance $\mathcal{I} \in \mathcal{H}$. We assume any policy has access to the correct marginal probability $\Pr(y \mid x, \mathcal{D})$ where $\mathcal{D}$ may contain current observations and/or some "lookahead" points, and we limit the lookahead amount to be $d$ since an optimal policy operates under such condition.

**Upper bound of an optimal policy.** With unlimited computational power, a policy can first compute the marginal probability of an arbitrary fixed clump point, conditioning on every possible subset of the isolated points of size $d$ with labels all equal to 1. This set of $\mathcal{O}(n^d)$ inference calls will reveal the location of the secret set, since exactly those points that could change the marginal probability of any fixed clump point are the secret points. Then the policy could query the identified positive clump and is guaranteed to achieve the target $T$ in time $T$. So the total cost of an optimal policy is upper bounded by $|\mathcal{S}| + T$, i.e.,

$$\text{OPT} < mdc + T = \frac{4\epsilon}{1 - 5\epsilon} n^{2\epsilon} \log n + n^{2\epsilon} = \mathcal{O}(n^{2\epsilon}). \tag{2}$$

In our asymptotic notations, all log factors will be omitted.

**Lower bound of any policy with limited computational power**. Fix a policy $\mathcal{A}$. Our goal is to show that with $o\left(n^{n^\epsilon}\right)$ inference calls, the expected cost of $\mathcal{A}$ is lower bounded by $\Omega\left(n^{3\epsilon}\right)$. The key is to bound the probability of $\mathcal{A}$ revealing the secret set throughout its execution. By Observation 1 and 2, $\mathcal{A}$ can make inference calls $Pr(y \mid x, \mathcal{D})$ to distinguish points in $\mathcal{S}$ from those in $\mathcal{R}$ only when $|\mathcal{D} \cap S| \geq d$. Suppose that before the $i$th inference call, $\mathcal{A}$ has no information about $\mathcal{S}$. Then the chance of $\mathcal{A}$ choosing a set $\mathcal{D}$ such that $|\mathcal{D} \cap S| \geq d$ is no better than that of a random selection from $n' = n - 2^m T = n - n^{6\epsilon} = \Theta(n)$ points. Since our goal is to prove a lower bound of the cost in the order of $\Omega(n^{3\epsilon})$, we allow $\mathcal{A}$ to make inference calls $\Pr(y \mid x, \mathcal{D})$ with $|\mathcal{D}| \leq n^{3\epsilon}$. Note this is much larger than the lookahead limit $d$. We can upper bound the probability of $\mathcal{A}$ choosing a dataset $\mathcal{D}$ such that $|\mathcal{D} \cap S| \geq d$, by counting how many subsets would contain at least $d$ points from $\mathcal{S}$, among all subsets of the $n'$ points of size at most $\beta = n^{3\epsilon}$:

$$\Pr\left(|\mathcal{D} \cap S| \geq d\right) \leq \frac{\binom{mdc}{d} \binom{n'-d}{\beta-d}}{\binom{n'}{\beta}} \tag{3}$$

$$< \left( \frac{\beta m d c}{n'} \right)^d \tag{4}$$

$$= \mathcal{O} \left( \frac{1}{n^{n^\epsilon}} \right) \tag{5}$$

With $\alpha$ such inference calls, the probability $p_h$ of at least one of them hitting the secret set can be union bounded by

$$p_h < \mathcal{O} \left( \frac{\alpha}{n^{n^\epsilon}} \right). \tag{6}$$

Hence for any $\alpha \leq n^{n^\epsilon - \delta}$ (for any positive constant $\delta$),

$$p_h < \mathcal{O} \left( \frac{1}{n^\delta} \right). \tag{7}$$

If $\mathcal{A}$ ever hits the secret set $\mathcal{S}$, we simply assume it will find all $T$ positives with zero cost. If not, then $\mathcal{A}$ can do no better than random selection, and its expected cost is $T/p_s$ (note $p_s > p_c$, and querying $n^{3\epsilon}$ clump points would not make the remaining clump points' probabilities higher than $p_s$). So the overall expected cost is lower bounded by $p_h \cdot 0 + (1 - p_h) \cdot \frac{T}{p_s} = \Theta(n^{3\epsilon})$. Here we used the fact that $p_s = 1 - \frac{1}{2^{1/c}} = \Theta(\frac{1}{c})$, which is easy to verify by L'Hôpital's rule.

Therefore,

$$\frac{\mathbb{E}[\text{cost}_\mathcal{A}]}{\text{OPT}} > \frac{\Omega(n^{3\epsilon})}{\mathcal{O}(n^{2\epsilon})} = \Omega \left( n^\epsilon \right). \tag{8}$$

That is, any policy $\mathcal{A}$ for cost effective active search with $\alpha = o(n^{n^\epsilon})$ inference calls would have expected cost at least $\Omega(n^\epsilon)$ times more than the optimal cost. The proof holds for $\epsilon$ such that $n' = n - n^{6\epsilon} = \Theta(n)$ indicating $6\epsilon < 1$, and for (5) to hold, we have $5\epsilon < 1$. We can set $\epsilon = 0.16$, which is less than $1/6$.

$\square$

**Remark 1.** *The parameters of the constructed instance are set to satisfy the following constraints:*

- *Make* OPT *linear in $T$, which means $T = \Omega(|S|)$.*

- *Our goal is to prove an $\Omega(n^\epsilon)$ bound, so the probability of the secret points $p_s = 1 - \frac{1}{2^{1/c}} = \Theta(1/c)$ should be $O(\frac{1}{n^\epsilon})$. This is because* OPT *is set to be linear in $T$ and the cost upper bound of a uniform random policy $T/p_s$ should be at least $\Omega(n^\epsilon) \cdot T$.*

- *Make the probability of secret points larger than that of the clump points, i.e., $p_s > p_c$, otherwise the cost upper bound would be $T/p_c$. So we have $\frac{1}{n^\epsilon} > \frac{1}{2^m}$. That makes $m > \epsilon \log n$.*

- *The number of clump points should be less than the total number of points. That is, $n > T \cdot 2^m$, which leads to $m < (1 - 2\epsilon) \log n$.*

- *$d$ controls the scale of the computational complexity bound and lookahead limit, the larger the tighter would be the bound.*

## 2 Example mass functions of negative Poisson binomial distribution

Figure 2 shows some examples of negative Poisson binomial distributions using the posterior marginal probabilities shown in 2a. We sort the probabilities in decreasing order since this is obviously the order that leads to minimum $\mathbb{E}[m]$, which is what we care in CEAS.

## 3 All results of materials and drug discovery

In the main paper, we only showed the results for ENS and CEAS with their best parameters. Table 1 and 2 show the full set of results for all tested policies. Note we did not test CEAS-50 and CEAS-0.5 for drug data since we expect them to be worse, according to Table 2.

Figure 3 show the cost curves for each individual drug discovery datasets. The average of the 9 plots is shown in the main text.

Figure 2: Illustration of a probability vector $[p_1, p_2, \ldots, p_n]$ and the corresponding probability mass functions of NPB distribution for different $r$. Left: the top 1500 posterior marginal probabilities after conditioning on 100 positive and 100 negative points (randomly selected); probabilities are computed using a $k$-nn model (with $k = 50$) on the CiteSeer dataset; middle: $r = 50$; right: $r = 200$.

Table 1: Results for all tested policies for the materials discovery dataset.

|           | 50   | 100   | 200   | 300   | 400   | 500   | 1000   | 1500   | average |
|-----------|------|-------|-------|-------|-------|-------|--------|--------|---------|
| GREEDY    | 84.5 | 175.0 | 347.7 | 522.5 | 721.8 | 924.1 | 2025.9 | 2981.7 | 972.9   |
| TWO-STEP  | 86.0 | 179.1 | 349.0 | 533.2 | 735.0 | 938.1 | 1973.1 | 3019.4 | 976.6   |
| ENS-10    | *81.7* | 167.8 | 339.2 | *520.4* | 721.9 | 939.8 | 1896.1 | 2836.5 | 937.9   |
| ENS-30    | *78.6* | 164.0 | *335.8* | 515.2 | 724.7 | 927.4 | 1795.8 | *2799.3* | 917.6   |
| ENS-50    | **78.0** | 162.5 | *329.6* | *517.0* | 729.6 | 926.6 | 1793.0 | 2812.2 | 918.6   |
| ENS-70    | *81.5* | 165.1 | *337.7* | 524.6 | 720.0 | 910.0 | 1790.6 | *2757.0* | 910.8   |
| ENS-0.1   | 84.2 | 177.2 | 343.2 | 520.6 | 717.7 | 946.3 | 1804.7 | *2765.1* | 919.9   |
| ENS-0.3   | *83.6* | 171.1 | 340.2 | 518.2 | *689.5* | 917.1 | 1815.9 | *2739.4* | 909.4   |
| ENS-0.5   | 82.3 | 162.7 | *335.4* | 535.3 | *693.2* | *897.9* | 1812.8 | *2736.4* | 907.0   |
| ENS-0.7   | *80.2* | 167.8 | **328.4** | *509.7* | 708.6 | *887.4* | 1798.5 | *2773.6* | 906.8   |
| ENCES-10  | 87.3 | 173.8 | 345.0 | 518.3 | 719.0 | 930.0 | 1825.5 | 2886.7 | 935.7   |
| ENCES-20  | 88.8 | 167.5 | *335.4* | 518.5 | 715.1 | 925.0 | 1779.7 | *2761.6* | 911.4   |
| ENCES-30  | 88.4 | 164.1 | *332.7* | *511.5* | 703.7 | *898.4* | **1734.7** | 2768.2 | *900.2* |
| ENCES-40  | *79.3* | **154.6** | *330.3* | 523.8 | 713.2 | 910.0 | *1748.8* | 2757.6 | *902.2* |
| ENCES-50  | 85.2 | *156.4* | *330.8* | *518.4* | *701.4* | *885.7* | *1745.4* | 2753.2 | *897.1* |
| ENCES-0.1 | 86.2 | 173.9 | 341.5 | 517.0 | 716.1 | 945.5 | 1844.1 | *2775.3* | 925.0   |
| ENCES-0.2 | 84.5 | 167.6 | *334.9* | **506.7** | 720.9 | 919.5 | 1845.1 | 2805.7 | 923.1   |
| ENCES-0.3 | 85.5 | 170.3 | *333.6* | *524.2* | 709.1 | *884.5* | 1823.6 | *2767.0* | 912.2   |
| ENCES-0.4 | 83.4 | *158.6* | *331.2* | 532.0 | 704.5 | **881.4** | 1797.0 | 2808.3 | 912.0   |
| ENCES-0.5 | 87.1 | 164.3 | *336.9* | *513.7* | **683.9** | *891.9* | *1774.1* | **2724.0** | **897.0** |

Table 2: Results for all tested policies for the nine drug discovery datasets.

|  | 50 | 100 | 150 | 200 | average |
|---|---|---|---|---|---|
| GREEDY | 215.7 | 414.4 | 503.2 | 587.4 | 430.2 |
| TWO-STEP | 71.7 | 156.0 | 243.2 | 322.4 | 198.4 |
| ENS-10 | *59.3* | 133.0 | 211.9 | 291.4 | 173.9 |
| ENS-30 | *58.8* | 134.9 | 208.3 | 283.3 | 171.3 |
| ENS-50 | *58.6* | 137.3 | 205.1 | 286.3 | 171.8 |
| ENS-70 | 80.2 | 197.4 | 207.3 | 288.8 | 193.5 |
| ENS-0.1 | 61.3 | 142.2 | 219.1 | 297.5 | 180.1 |
| ENS-0.3 | 59.5 | 133.0 | 215.3 | 292.8 | 175.2 |
| ENS-0.5 | *59.2* | 132.6 | 215.0 | 287.9 | 173.7 |
| ENS-0.7 | *59.1* | 132.8 | 212.0 | 284.2 | 172.0 |
| ENCES-10 | *57.3* | 119.5 | 196.2 | 273.6 | 161.7 |
| ENCES-20 | **56.3** | **112.7** | **184.5** | **255.1** | **152.2** |
| ENCES-30 | 75.1 | *128.4* | *196.5* | *261.5* | 165.4 |
| ENCES-40 | 102.4 | 165.9 | 221.5 | *282.3* | 193.0 |
| ENCES-0.1 | *58.2* | 197.3 | 195.0 | 271.4 | 180.5 |
| ENCES-0.2 | *72.9* | 116.0 | 194.8 | 298.9 | 170.7 |
| ENCES-0.3 | *57.1* | 121.5 | 267.6 | 318.9 | 191.3 |
| ENCES-0.4 | 56.8 | *134.4* | 275.5 | 319.8 | 196.6 |

Figure 3: Average cost versus the number of positives found for 9 drug discovery datasets. The total number of positives are 553, 378, 506, 1023, 218, 916, 1024, 431, 255, respectively.

Table 3: Average pruning rate across all iterations in all experiments for the reported ENCES policies.

| BMGs | | drug discovery | |
|---|---|---|---|
| ENCES-50 | ENCES-0.5 | ENCES-20 | ENCES-0.2 |
| 98.64% | 98.01% | 99.03% | 99.06% |

## 4   Implementation and pruning

To enable fast update of the posterior probabilities, we use $k$ nearest neighbor ($k$-nn) model, following Garnett et al. [2012] and Jiang et al. [2017, 2018]. We briefly describe the model again. Intuitively, the posterior probability of a point is computed by counting the proportion of observed positive points in its $k$ nearest neighbors. Formally, let $\mathrm{N}(x)$ denote the set of $k$ nearest neighbors of $x$, and $\mathrm{LN}(x) \subseteq \mathrm{N}(x)$ be the possibly empty subset that is currently labeled. Then the posterior probability of $x$ conditioned on $\mathcal{D}$ is

$$\Pr(y = 1 \mid x, \mathcal{D}) = \frac{\gamma + \sum_{x' \in \mathrm{LN}(x)} y'}{1 + |\mathrm{LN}(x)|}. \tag{9}$$

Here $\gamma$ is a constant accounting for "pseudocount" of the positives. We set it to our prior belief of the proportion of positives in the pool. So when there are no observations in its $k$ nearest neighbors, we use the prior belief. For BMGs data, we set $\gamma = 0.05$. For the drug discovery datasets, we set $\gamma = 0.001$.

When conditioning on a new point $x$, only those points having $x$ in its $k$ nearest neighbors need to be updated. Assume conditioning on one point changes at most $\tilde{k}$ probabilities ($\tilde{k}$ is roughly $\mathcal{O}(k)$), then use the implementation tricks in Jiang et al. [2017], our policy can be computed in $\mathcal{O}\left(n \log n + n\left(\tilde{k} \log \tilde{k} + \mathbb{E}[m^-]\right)\right)$ (note under the $k$-nn model, $E[m^-]$ is always no less than $\mathbb{E}[m^+]$).

To further improve the computational efficiency, we develop similar pruning techniques as for policies in the budgeted case [Garnett et al., 2012, Jiang et al., 2017, 2018]. Assume we have upper bounds $\hat{p}$ of the probabilities after one additional positive observation: that is $\forall x \in \mathcal{X} \setminus \mathcal{D}_i$,

$$\hat{p}(x) \geq \Pr(y \mid x, \mathcal{D}_i, x', y' = 1), \forall x' \in \mathcal{X}. \tag{10}$$

We can compute a lower bound of $\mathbb{E}[m^+]$ using $\hat{p}$. If we also assume observing a negative does not increase the probability of any other point (the $k$-nn model satisfies this condition), then a natural probability upper bound after observing a negative point is simply the current probability

$$\Pr(y \mid x, \mathcal{D}_i) \geq \Pr(y \mid x, \mathcal{D}_i, x', y' = 0), \tag{11}$$

and we can compute a lower bound of $\mathbb{E}[m^-]$ using $\Pr(y \mid x, \mathcal{D}_i)$. Combining both lower bounds we get a lower bound of $f(x), \forall x \in \mathcal{X} \setminus \mathcal{D}_i$. It is easy to see the bound is tighter with smaller $r$. This bound can be used to prune points in a similar fashion as in [Jiang et al., 2018].

To find the point $x^* = \arg \min_x f(x)$, we evaluate the candidate points in increasing order of the lower bound, and maintain the minimum of currently evaluated points as an upper bound of $\min f(x)$; we can stop whenever the lower bound becomes greater than the upper bound. We will show that often only a very small percentage (e.g. 1%) of the candidate points need to be evaluated in each iteration.

**Pruning results**. We also show the effectiveness of our pruning technique in Table 3. The second row indicates the policy as reported in the main text. The third row shows the average percentage of pruned points over all candidates in each iteration; the average is taken over all iterations and all experiments. We see the pruning is very effective on all datasets; most of the time only 1% of the points need to be evaluated in each iteration.

## Footnotes

[1] here dividing by $(1 - 5\epsilon)$ is not essential; only for the purpose of getting a simpler formula in our theorem.