[Reviews · NeurIPS 2019]

Reviewer 1



The paper considers a Bayesian decision theoretic formulation of the problem of minimizing the number of queries to identify the desired number of positive instances (instances with positive labels), given a probabilistic model of the labels in the dataset. This formulation is motivated by the material and drug discovery problems. The problem is properly formulated and contrasted with the recently suggested budgeted-learning setting, where the goal is to identify the largest number of positive instances given a fixed budget on queries. Further the authors show that the optimal Bayesian policy is hard to compute and hard to approximate. However, further assuming certain conditional independence the policy can be approximated efficiently using the negative-poisson-binomial distribution, for which the authors propose computationally-cheap expectation estimates.The resulting policy is compared to several other alternatives, and it is shown to obtain overall superior performance in both material discovery and drug discovery datasets. Quality: Most of the material presented in the paper is sound and the experimental results are convincing. The proof for the main theoretical statement is provided in the supplementary material, but I didn’t check it. There are however some gaps that are not completely satisfactory in the main text. Specifically, it is about the approximation of the expected value of the NPB distribution discussed in section 4.2. A single set of experimental results suggesting that the proposed estimator ACCU’ is accurate is insufficient in my opinion to use, given that its accuracy is central to the policy choice. Obtaining some sort of bounds is important. On the other hand, it is also not completely clear to me why the \epsilon-DP is computationally expensive in the first place. After all, a single evaluation of the instance is assumed to be expensive (time, resources, etc.), so spending more time to accurately getting the policy should not be a bottleneck. Another option would be estimating the expected value by sampling. There are lots of exponential tail bounds that give solid guarantees, wouldn’t this work? Regarding the experiments, I don’t see enough information about the details of the posterior probability model, beyond a short mentioning that it is based on a k-nn. A detailed model should be explained somewhere in the text (main or supplementary). Furthermore, there are no details regarding how the ENS is modified to match the CAES setting in the experimental setup. This is particularly interesting as it is important to know if there are relatively easy ways to map one problem to the other, despite the opposing arguments put forward in the introduction. On that note, I would suggest to revise this argument (the third paragraph in the into). Why is the problem not one-to-one? Does it even matter? Originality: As far as I know this work is original, starting from the suitable problem formulation and down to the algorithmic implementation. It certainly adapts ideas from prior work but they are sufficiently different. Clarity: The paper is overall well written. Significance: I believe that the work presented is sufficiently significant and will be beneficial for suitable search problems that are not very unique in the general scientific community. It provides a new problem formulation that is more suitable to those search problems, the algorithms are implemented, discussed and compared on two interesting real datasets. Also, the difficulty of approximating an optimal Bayesian policy for a general posterior model is an important (though not surprising) theoretical result. Misc: Line 129: What is “n” in O(n^t)?

Reviewer 2



*Originality* The abstract claims that the current submission is the first one to systematically model the active search problem as a Bayesian formulation and the resulting algorithm outscores prior work. However, the theoretical model in paper is very limited. The authors assume a fixed prior and derive a Bellman equation. There is some novelty in the choice of the state and to include number of queries remaining in the state, but these are all standard tricks in the MDP literature. Furthermore, at the end the algorithm presented is a heuristic one. The authors assume -- as the prior work -- that the distribution of the state of their MDP does not change with the outcomes of the currently considered points. This decoupling assumption relieves the algorithm of the complexity of computing these posteriors for each point at each step. They simply assume a fixed distribution for the state with updated parameters. Specifically, they assume a negative Poisson Binomial distribution (which they have defined in the paper and is perhaps new). This is itself a heuristic and places the current submission at par with prior heuristic search methods. *Quality* For me, this work is mainly experimental. The authors have not compared with the large body of theoretical work on active search, which perhaps does not lead to algorithms as efficient as the authors desire, but is theoretically more rich. The authors should have at least clarified why that literature is not relevant here. Further, as described above, the only theory developed in the paper is the Bellman equation for an appropriate state function, which is interesting but not comprehensive. Thus, I think the main contribution of the paper is a heuristic algorithm which fares better than some prior work on specific data sets. For experiments, the authors considered two use-cases of drug discovery and material discovery and compared many existing methods. However, I found the improvement reported in Figure 1(b) rather marginal. But the numbers reported in Table 2 are perhaps acceptable as a benchmark for convincing numerical improvement. I am not an expert on this point. *Clarity* The presentation is very confusing. Even the problem formulation was never concretely laid down and it took me several readings to reconstruct the problem. What Bayesian formulation are the authors referring to? Is the the knowledge of the exact posterior assumed or only within a family? Is the cost the expected number of queries or you want a confidence bound? All these questions are eventually clarified but never explicitly. I would have preferred a complete formulation at the outset. Also, it was never clarified which distribution (what parameters for NPB) are used in the first step of iteration. I understand the posterior updates from there on, but what do you start with? I assume all p_is are set as 1/2 first. Overall, I think it is an interesting paper but below par for NeurIPS publication. [Update after rebuttal] The authors have not really clarified my two important points: How is their assumption of distribution justified (is it the least-favorable one and matches some minimax lower bound?) and how is their decoupling assumption justified? However, reading other reviews I feel that these are not the main concerns for people working in this area. So, I can't increase the score, but I can reduce my level of confidence to 2. Perhaps the area chair can take a look and decide.

Reviewer 3



In this paper the authors investigate the following cost-effective active search problem: the goal is to find a given number of positive points which minimize the label cost. Here they used the Bayesian decision model. They show a strong hardness result: any algorithm with complexity O(n^{n^0.16}) cannot approximate the optimal cost with in \Omega(n^0.16). Previously there is an upper bound O(1/\sqrt{log n}) for the maximization version of the problem. They also give an approximation algorithm for the minimization problem as well as some experiment under different scenarios. Here are some detailed comments: The author claim that it results in an exponentially stronger bound (n^0.16 vs \sqrt{log n}). (Line 147) However, the two problem are different, one is minimization problem, one is maximization problem. There is no necessary connection between the approximation ratio for the min and the max problem. For example, for set cover problem, the approximation ratio for minimization version is O(log n), but the approximation for maximization version is 1-1/e. For the hardness result, what is the dependent on the parameter T? Because when T=O(1), there is some DP algorithm which can solve the problem in polynomial time. Line 110, “Let p(x)=p(y=1:x,D_i)”, here the notation of p(x) does not show the dependence on index i

Reviewer 4



The paper is well structured and is easy to follow for the most part. The authors proposed to approximate the expected remaining cost via the expectation of a negative Poisson binomial (NPB) distribution. The proposed nonmyopic algorithm, ENCES, aims at minimizing the approximated expectation of the corresponding NPB distribution. Note that this is an approximation because the conditional independence (CI) assumption does not always hold. Can the authors please justify, either empirically (on small toy dataset) or theoretically, how well the NPB expectation approximate the true expected cost? In addition to the query complexity, it will be useful to see the time/computational complexity of the competing algorithms, e.g., how much more expensive is ENCES than the (one-step) greedy algorithm? == post-rebuttal comments: Thanks a lot for the clarification on the conditional independence (CI) assumption. One comparison I would like to make is against variational inference, which also makes independence assumption for the approximate distribution -- however, VI tries to find the closest approximation in a parameterized family. In contrast, here the conditional independence may not hold and hence it's hard to provide any guarantees on the performance based on the proposed search heuristics. I think this is still a reasonable heuristic to run (and a good one given the experimental results), while there is still room for improvement on the theoretical aspects.

[Author Response · NeurIPS 2019]

We thank the reviewers for their detailed and insightful comments!

**R3**: In fact we have performed experiments using $\epsilon$-DP in place of ACCU'. This procedure is somewhat more expensive
but indeed still acceptable in terms of runtime. Our preliminary results showed that ACCU' led to almost identical
results, so we adopted this simpler and cheaper alternative for our experiments. We will add a note to this effect. We
also note that ACCU' can also be motivated from an intuitive perspective: if a coin $i$ contributes $p_i$ HEADS in expectation,
how many coins do we need to flip to attain the desired number? This intuitively corresponds to the cost of active
search. A more systematic theoretical and empirical study of ACCU' for NPB is an interesting topic on its own.

We have also investigated Monte Carlo (see `compute_negative_poisson_binomial_expectation_monte_carlo`
`.cpp` in the supplementary material under `code/min_cost/`). Empirically (and unsurprisingly), it was much more
expensive than $\epsilon$-DP to achieve the same error level.

The specific $k$-nn probability model is described in Eq. (7) of Garnett et al. (2012) (line 327). Basically, the prior
probability of each point being positive is the estimated marginal probability (e.g., 0.01). After each observation, these
probabilities are updated by counting the proportion of positives in each point's $k$ nearest neighbors, smoothed by the
prior. This is a simple but effective model for active search. We will add more information to the appendix.

The only modification needed to adapt ENS to the cost-sensitive setting is appropriately specifying the "budget," as
described lines 258–265 in the main text. The two settings are not directly related, as one is a maximum coverage
problem and the other is a covering problem. However, the two problems are related in that one can be thought of as the
dual of the other. This connection is perhaps why ENS is such a strong baseline for the CEAS setting.

In line 129, $n$ is the number of candidate points to choose from.

**R5**: We strongly disagree that our "main contribution is a heuristic algorithm." We have (1) introduced a new
optimization problem extending those considered in the literature, opening the potential for a new line of work and (2)
established the optimal policy for this problem. This policy is computationally intractable; however, we (3) provide a
fast and empirically strong approximate policy, guided by the optimal policy. Finally, we (4) provide interesting lower
bounds on the approximation ratio for this problem. Although (3) is a heurisitc algorithm, our theoretical work (1, 2, 4)
shows that heuristic algorithms are the best we can hope for due to the inherent hardness of the underlying problem (4).

"The authors have not compared with the large body of theoretical work on active search." Again, we disagree. We have
compared with both the most relevant work on active search (Garnett, et al. (2012) and Jiang, et al. (2017, 2018a)),
as well as with work from the stochastic submodular optimization literature. It is difficult to respond further as you
have not identified any *particular* missing work. If the reviewer could cite the work they are referring to, we would be
delighted to include it in our discussion.

"The improvement reported in Figure 1(b) is marginal." This is perhaps due to the visual contrast with the gap between
one-step and two-step policies. Achieving a 5–10% improvement is exciting for applications where each experiment
is costly. Further, our algorithm is consistently over 50% better than the popular greedy heuristic on drug discovery
datasets, a massive reduction in cost.

The prior marginal distribution of points being positive is Bernoulli with a constant parameter $p$, the estimated ratio of
positives in the pool (e.g., $p = 0.01$).

**R6**: $n^{0.16}$ vs $\sqrt{\log n}$: We absolutely agree with the reviewer that the two problems are different, analogous to set cover
versus maximum coverage. It is not surprising that the two problems have different complexity. What we believe is
interesting is that this paper formally establishes that cost effective active search is much harder to approximate than
known bounds on active search. Beforehand, it was not obvious this problem should have a polynomial lower bound on
the approximation ratio, and we suspected initially that it would be poly-logarithmic (such as in set cover).

Dependence on $T$: In the proof of Theorem 1, we used a construction where $T = n^{2\epsilon}$. In practice, the target number of
positives usually grows as the total number of points. Besides, theoretically it is not very meaningful to assume constant
$T$, since then even a random policy would have constant expected cost of $T/p$, where $p$ is the ratio of positives.

**R7**: "how well the NPB expectation approximates the true expected cost": this is a great question. In short, it is provably
infeasible to even approximate the true expected cost and we cannot expect *any* computationally fast algorithm to
approximate the true expected cost (see Theorem 1 at line 135). We show this by proving the problem has a strong
lower bound on the approximation ratio even if super polynomial time is allowed. Empirically, we cannot compute the
true expected cost for even dozens of points (recall the $\mathcal{O}(n^t)$ complexity at line 129). This lower bound can perhaps be
circumvented if some degree of conditional independence holds or some other useful structure in the probability model.
Determining which probability distributions result in the ability to approximate the true expected cost is an exciting line
of future research.

[Meta-Review · NeurIPS 2019]

Most of the reviewers were in favor of accepting. The main reservation expressed by the reviewers was that the assumptions of the theory are unrealistic, and so the theoretical contributions shouldn't be viewed as particularly significant in themselves. Specifically, the algorithm is motivated by some theory that is based on a conditional independence assumption that might not typically be satisfied. However, they showed (via a computational hardness result) that some kind of assumption is necessary for a solution to be tractable at all. So, since the resulting method seems reasonable and exhibits good empirical performance, the reviewers and I were mostly willing to look past this assumption.